# HPLC Fingerprint Analysis of *Cibotii rhizoma* from Different Regions and Identification of Common Peaks by LC-MS

**DOI:** 10.3390/ph17030313

**Published:** 2024-02-28

**Authors:** Zhongjing Guo, Yu Duan, Zhimin Zhao, Depo Yang, Xinjun Xu

**Affiliations:** School of Pharmaceutical Sciences, Sun Yat-sen University, Guangzhou 510006, China; guozhongjing2020@163.com (Z.G.); 15666558496@163.com (Y.D.); zhaozhm2@mail.sysu.edu.cn (Z.Z.); lssydp@mail.sysu.edu.cn (D.Y.)

**Keywords:** *Cibotii rhizoma*, HPLC, fingerprint analysis, hierarchical cluster analysis, orthogonal partial least squares discriminant analysis, quality evaluation

## Abstract

To establish the fingerprint of *Cibotii rhizoma* using high-performance liquid chromatography (HPLC) and evaluate the quality of *Cibotii rhizoma* from different regions using chemometrics to identify the potential quality markers, thirteen batches of *Cibotii rhizoma* samples were analyzed. the similarity evaluation system of TCM chromatographic fingerprint similarity evaluation was used to confirm common peaks. The SPSS 27 software was used for hierarchical cluster analysis (HCA), and SIMCA 14.1 software was used for principal component analysis (PCA) and orthogonal partial least squares discriminant analysis (OPLS-DA). Moreover, a batch of *Cibotii rhizoma* was selected for LC-MS analysis and speculated on 15 common components. HPLC fingerprint were established, 15 common peaks were matched, two chromatographic peaks were identified using standard substances (protocatechuic acid and protocatechuic aldehyde), and 13 common components were inferred through liquid chromatograph-mass spectrometer (LC-MS). The 13 batches of the samples showed good similarities (>0.910). The results of HCA, PCA and OPLS-DA showed that 13 batches of samples were divided into three groups, and different markers were selected. The method is simple, rapid and reproducible, and can provide a reference for the overall quality evaluation of *Cibotii rhizoma*.

## 1. Introduction

*Cibotii rhizoma* is derived from the rhizome of the Dicksoniaceae plant *Cibotium barometz* (L.) J.Sm., which has the effects of dispelling wind and dampness, nourishing liver and kidney, and strengthening waist and knee, and can be used for rheumatism and pain, waist and knee pain, lower limb weakness and other diseases [1]. *Cibotium barometz* (L.) J.Sm. is widely distributed in subtropical and tropical areas, including China, Thailand, India, Malaysia and Indonesia. In China, *Cibotium barometz* (L.) J.Sm. is mainly distributed in the provinces of Yunnan, Guizhou, Sichuan, Guangxi, Guangdong, Fujian, Taiwan, Hainan, Zhejiang, Jiangxi and Hunan [2], and it is one of the important traditional Chinese medicines exported from China [3]. The chemical constituents of *Cibotii rhizoma* mainly contain sugars and glycosides, aromatic species, volatile oils, ferns and flavonoids. And the plant exhibits many pharmacological effects such as antioxidant, anti-inflammatory, antiosteoporosis and anti-osteoarthritis [4]. Due to different factors such as humidity, temperature and soil in different places, the content of effective components of medical materials is affected, resulting in different efficacy rates of medical materials from different regions. *Cibotium barometz* (L.) J.Sm. is an indicator plant for acidic soil and has strict environmental requirements. Some researchers have found that *Cibotii rhizoma* currently circulating in the market is wild and has not been cultivated on a large scale [5]. The quality of medical materials from the wild is different, and the chemical content of *Cibotii rhizoma* from different regions is unknown. In order to ensure the quality of *Cibotii rhizoma* sold on the market, it is necessary to explore the chemical content of *Cibotii rhizoma* from different regions, providing a basis for quality evaluation and control of *Cibotii rhizoma*.

Traditional Chinese medicine fingerprint technology is an effective method to evaluate the advantages and disadvantages of traditional Chinese medicine, identify authenticity, distinguish species and ensure its consistency and stability, and it is also one of the effective means to control the quality of natural medical materials [6]. Chromatographic fingerprint analysis has become a consensus for the quality assessment of herbal medicine by the American Food and Drug Administration (FDA) [7] and the European Medicines Agency (EMA) [8]. In the past decades, fingerprint analysis has been widely used for authenticating, qualitatively evaluating and tracing the origin of botanical products due to its characteristics of simplicity, convenience and efficiency [9]. However, the amount of information displayed by a fingerprint is too large, and only the similarity of fingerprints can be calculated, which is not conducive to statistical analysis. In recent years, as an important part of chemometrics, pattern recognition technology has become an important mathematical method to extract the characteristics of complex chemical fingerprints of traditional Chinese medicine, with good prediction accuracy and wide applicability [10]. The combination of the two can quickly screen out the main different components and achieve the purpose of distinguishing samples from different regions.

At present, studies on the characteristics of *Cibotii rhizoma* include the fingerprint of *Cibotii rhizoma* polysaccharide [11], the fingerprint of Stir-baked *Cibotii rhizoma* standard decoction [12] and the pre-processing of *Cibotii rhizoma* fingerprint [13], but there are very few reports on the HPLC fingerprint analysis based on soluble phenolic acid in *Cibotii rhizoma* at present. Protocatechuic acid and protocatechuic aldehyde are two important soluble phenolic acids. Protocatechuic acid is a compound of multidirectional biologicalactivity, i.e., anti-inflammatory, antioxidative, antibacterial, antiviral, analgesic, antiatherosclerotic, hepatoprotective and antineopla [14]. Protocatechuic aldehyde has been reported to possess anti-inflammatory, antiproliferative, and antioxidant properties in various in vivo and in vitro experiments [15]. In this work, fingerprints of 13 batches of *Cibotii rhizoma* samples from seven producing areas are established using HPLC, and 13 common components are inferred through mass spectrometry. The overall quality is evaluated by combining chemical pattern recognition, and the different components of different producing areas are screened so as to understand the relationship between *Cibotii rhizoma* quality and origin and provide references for the improvement of *Cibotii rhizoma* quality control.

## 2. Results and Discussion

### 2.1. Optimization of HPLC Conditions

This experiment investigated the effects of four extraction solvents (methanol, methanol-1% glacial acetic acid (70:30), ethyl acetate, and n-butanol) on the number, resolution, and shape of chromatographic peaks in *Cibotii rhizoma* samples. The results showed that methanol-1% glacial acetic acid (70:30) used as the extraction solvent provided the best extraction results. In the study of chromatographic conditions, three mobile phase systems of methanol-0.1% glacial acetic acid, methanol-1% glacial acetic acid and acetonitrile 1% glacial acetic acid were selected for investigation, as well as Ultimate AQ-C_18_ column (4.6 mm × 250 mm, 5 μm) and JADE-PAK ODS-AQ C_18_ column (4.6 mm × 250 mm, 5 μm). The acetonitrile (A) −1% glacial acetic acid solution (B) solvent was finally determined as the mobile phase. When using the Ultimate AQ-C_18_ (4.6 mm × 250 mm, 5 μm) column, the system baseline was stable and chromatographic peak separation was good under this condition. When the wavelength was 260 nm, there were more fingerprint peaks and the response value was higher, so the detection wavelength was set at 260 nm. The optimal chromatographic conditions were used as described in Section 2.3.

### 2.2. Method Validation

The precision, stability and repeatability were assessed by detecting relative retention time (RRT) and relative peak area (RPA) of the compounds, respectively. The RSDs of the RRT and RPA were less than 0.4% and less than 2.2%, respectively, which demonstrated that the precision of the equipment was good. The RSDs of the RRT and RPA were less than 0.6% and less than 3.5%, respectively, demonstrating that all of the analytes are stable after their extraction from *Cibotii rhizoma* for at least 24 h. The RSDs of the RRT and RPA were less than 1.2% and less than 5.0%, respectively, which demonstrated that the repeatability of the method was good. All the results of the method validation tests demonstrated that the proposed method was reliable and valid.

### 2.3. Establishment of the HPLC Fingerprint

Thirteen batches of *Cibotii rhizoma* sample solutions were prepared according to the method described in Section 3.2. Samples S1–S13 were injected according to the chromatographic conditions described in Section 3.3, and chromatograms were recorded. The fingerprints of 13 batches of *Cibotii rhizoma* were established by the software of Similarity Evaluation System for Chromatographic Fingerprint of Traditional Chinese Medicine [16,17,18]. The results of similarity evaluation are shown in Table 1. In the software, the reference fingerprint was obtained using the median method, and the time window width was set to 0.3 min. After multi-point correction and automatic matching, the HPLC superposition atlas and control fingerprint of 13 batches of *Cibotii rhizoma* were generated. A total of 15 common peaks were calibrated, and two of them were identified as protocatechuic acid (peak No. 5) and protocatechuic aldehyde (peak No. 10) through comparison with the reference standards. HPLC fingerprints for 13 batches of *Cibotii rhizoma* (A) and mixed standard chromatogram (B) are shown in Figure 1. Because of the good shape of the common peaks and good separation from each other, the sample of the S5 chromatogram (C) was chosen to display the separation of common peaks.

### 2.4. Similarity Evaluation

The characteristic mode of the HPLC fingerprint was used as the reference fingerprint for similarity evaluation. As shown in Table 1, the similarity between the 13 batches of *Cibotii rhizoma* samples and the reference fingerprint (R) ranged from 0.998 to 0.785. The similarity between S10 and R was the lowest (0.785). This indicated that there were great differences between S10 and other batches of *Cibotii rhizoma*. It indicated that the chemical composition of this batch of *Cibotii rhizoma* was very different from that of other batches, and the similarities of other batches of *Cibotii rhizoma* were above 0.9, indicating that the chemical composition of the 12 batches of samples were basically the same.

Peak No. 11 with the largest peak area, a moderate peak time and a relatively stable peak was used as the reference peak (S). The RRT and RPA of fifteen common fingerprint peaks were determined. The RRT and RPA of all the common characteristic peaks with respect to this reference peak were calculated; the RPA and RSDs are shown in Table 2. The RSDs of the relative retention time were low (less than 0.5%), which indicated that the peak time of common peaks in different samples was relatively stable; meanwhile, the RSDs of the relative peak area were high, indicating that the content of the same chemical component in the *Cibotii rhizoma* from different regions was significantly different. Using the external standard method, the contents of protocatechuic acid and protocatechuic aldehyde in each batch of samples were calculated; the results are shown in Table 3. It can be seen that the content of both components in each batch was relatively low, among which the content of protocatechuic acid in S10 was significantly higher than that in other batches, which is consistent with the similarity evaluation results.

### 2.5. Hierarchical Cluster Analysis (HCA)

The hierarchical cluster analysis of 13 batches of *Cibotii rhizoma* was performed using the IBM SPSS Statistics (Version 27.0) software based on the standardization of 15 common peak areas using the inter-group join method and the squared Euclidean distance as the measure of sample similarity; the dendrogram of HCA is shown in Figure 2. According to the results of HCA, the cross-over phenomenon existed in the *Cibotii rhizoma* collected from different regions, and the quality difference of *Cibotii rhizoma* may not have been associated with the origin. When the squared Euclidean distance was 10, the 13 batches of *Cibotii rhizoma* samples were divided into three groups (G1, G2, G3). G1 (including S1, S5, S6, S7 and S8) mainly comes from the provinces of Guangxi and Sichuan. G2 (including only S10) comes from the provinces of Jiangxi, showed the lowest similarity evaluation compared to other samples. G3 (including S2, S3, S4, S9, S11, S12 and S13) mainly comes from the provinces of Guangxi, Guizhou, Fujian and Guangdong. The HCA classification results were consistent with the similarity evaluation results. This denoted that HCA was an efficient way to identify *Cibotii rhizoma* from different regions [19].

### 2.6. Principal Component Analysis (PCA)

Using the peak areas of 15 common peaks from 13 batches of *Cibotii rhizoma* fingerprint as variables, the data were standardized using IBM SPSS Statistics (Version 27.0). Principal component analysis was performed on the standardized data. The results are shown in Table 4. The scree plot of the PCA is shown in Figure 3A. Combined with the mutation point of the scree plot, the three principal components were extracted for evaluation. Three principal components (PC1, PC2 and PC3) accounting for 46.430%, 26.763%, 11.182% of the total variance, respectively, were selected to represent the total variable information based on eigenvalue >1. The first three principal components occupied 84.375% of the total variable, which represented most of the raw data. The component matrix reflects the magnitude and direction of each variable’s contribution to the principal component. A larger value indicates a higher correlation with the principal component, while a negative value indicates a negative correlation with the principal component. From the 3D loadings plot of the PCA (Figure 3B), P3, P6, P7, P9, P11, P12, P13, P14 and P15 contributed the most to Principal Component 1, P1, P4, P5 and P8 contributed the most to Principal Component 2, P2 and P10 contributed the most to Principal Component 3 (P refers to the peak number). The first principal component has the largest characteristic value, and the weight value of its variables can reflect the correlation between chemical components and the quality of herbs to the greatest extent [20]. Unknown P3, P6, P7, P9, P11, P12, P13 and P14 make the greatest contribution to Principal Component 1, indicating that these components are important factors affecting the quality of *Cibotii rhizoma*.

Using the peak areas of 15 common peaks from 13 batches of *Cibotii rhizoma* fingerprint as variables, PCA analysis was performed using the SIMCA (Version 14.1) software. A PCA model was established, scaled by the equal variance method, and three principal components were extracted. Model interpretation rate parameter R^2^X = 0.899, and prediction ability Q^2^ = 0.627. The results showed that the extracted three principal components can explain 89.9% of the original variables, and the prediction ability of the model was 62.7%. The model had good variable interpretation ability and group prediction ability. The PCA scores of 13 batches of *Cibotii rhizoma* samples are shown in Figure 3B. The results showed that the samples were basically divided into three groups (G1, G2, G3). G1 includes S1, S5, S6, S7 and S8, G2 includes S10, and the other samples are in G3. The results were consistent with those of HCA.

### 2.7. Orthogonal Partial Least Squares Discriminant Analysis (OPLS-DA)

In order to further search for markers leading to the quality differences of *Cibotii rhizoma* from different regions, the 13 batches of samples were analyzed by orthogonal Partial least squares discriminant analysis (OPLS-DA). The peak areas of 15 common peaks were imported into the SIMCA (Version 14.1) software, and the feature scaling was carried out by the Pareto algorithm to establish the OPLS-DA model of 13 batches of samples divided into three groups. The score of OPLS-DA is shown in Figure 4A. The model interpretation rate parameters, R^2^X and R^2^Y, were 0.95 and 0.977, and the prediction ability parameter Q^2^ was 0.761, both greater than 0.5. It shows that the model is stable and reliable. The permutation test (*n* = 200) was used to verify the model, and test parameters R^2^ = (0, 0.804) and Q^2^ = (0, −1.2) are shown in Figure 4B. Both R^2^ and Q^2^ generated by random arrangement on the left were smaller than the original values on the right, indicating that the OPLS-DA model was well fitted.

The variable importance in projection (VIP) value is an important indicator for screening differential components. The higher the VIP value, the greater its impact on inter-group differences. With a VIP value greater than one as the threshold, seven common peaks were screened out, as shown in Figure 4C. The VIP characteristic values of P14, P8, P4, P15, P5 (protocatechuic acid), P12 and P11 were greater than one, so these peaks were the marker components that caused the differences between *Cibotii rhizoma* in different regions.

### 2.8. Identification of Main Chemical Components

The sample of *Cibotii rhizoma* is analyzed according to the chromatographic mass spectrometry conditions under Section 3.6 and the mass spectrometry peak map is obtained in negative ion mode. The results are shown in Figure 5. Based on the characteristic UV absorption wavelength, precise molecular weight provided by high-resolution mass spectrometry, and information on characteristic fragment ions, a total of 13 components were identified through literature review or comparison with standard samples, including nine glycoside components, three phenolic acid components, and one flavonoid component. The total ion chromatogram is shown in Figure 5A and the identification results are shown in Table 5.

Peak 5 showed the [M-H]^−^ ion at *m*/*z* 153.0242, by which its molecular formula was determined as C_7_H_6_O_4_. In the MS^2^ experiment, the fragment at *m*/*z* 109.0286 was corresponding to the loss of CO_2_. From the above analyses, this peak was deduced as protocatechuic acid by comparing with the literature [21] and confirmed by the reference substance, as shown in Figure 5B. Peak 6 was assigned the molecular formula of C_13_H_14_O_9_ by the [M-H]^−^ ion at *m*/*z* 313.0530. When the [M-H]^−^ ion was selected for MS/MS experiment, prolific fragmentation ions were obtained. The fragment ions at *m*/*z* 153.0221 and *m*/*z* 108.7908 were generated by the loss of C_6_H_8_O_5_ and CO_2_. From the above analysis, this peak was tentatively deduced as cyathenosin A [22], as shown in Figure 5C. Peak 9 displayed the [M-H]^−^ ion at *m*/*z* 329.0871, corresponding to the molecular formula of C_14_H_18_O_9_. The MS^2^ fragments at *m*/*z* 167.0440 and *m*/*z* 123.0285 were corresponding to the loss of C_6_H_10_O_5_ and CO_2_; this peak was determined as vanillic acid 4-O-β-D-glucopyranoside [23], as shown in Figure 5D. Peak 11 displayed the [M-H]^−^ ion at *m*/*z* 341.0865, corresponding to the molecular formula of C_15_H_18_O_9_. The MS^2^ fragments at *m*/*z* 281.0588 and *m*/*z* 179.0324 were corresponding to the loss of COOH and C_6_H_11_O_5_; this peak was determined as 1-caffeoyl-β-D-glucose [24], as shown in Figure 5E. Peak 14 showed the [M-H]^−^ ion at *m*/*z* 411.0959, by which its molecular formula was determined as C_18_H_20_O_11_. In the MS^2^ experiment, the fragment at *m*/*z* 315.0666 corresponded to the loss of C_5_H_5_O_2_. From the above analyses, this peak was deduced as cibotiumbaroside G [25], as shown in Figure 5F.

### 2.9. Advantage of the Study

Xu [11] studied the fingerprint of *Cibotii rhizoma* polysaccharide accounting, but the separation of the obtained polysaccharide HPCE (high-performance capillary electrophoresis) spectrum was not good enough. Huang [12] studied the fingerprint of Stir-baked *Cibotii rhizoma* standard decoction, but there were certain differences in the chemical composition between the decocted medicinal materials and the crude drug which were not suitable for quality control of the crude drug. Moreover, this study did not identify the common peaks obtained. Chen [24] used ultra-high performance liquid chromatography tandem quadrupole time-of-flight mass spectrometry (UPLC-Q-TOF/MS) to characterize the chemical components in *Cibotii rhizoma*, identifying a total of 26 chemical components, which is beneficial for further research on the chemical components in *Cibotii rhizoma*. The chromatographic method established in this study was suitable for the separation of various components in *Cibotii rhizoma*, and the obtained chromatograms have good shape of peaks. The use of the crude drug has more reference value for quality control. Fingerprints were established, which can provide a certain reference for the overall evaluation of *Cibotii rhizoma*. By combining the methods of chemometrics, the 15 common peaks obtained were analyzed, and the indicator substances that affect the quality of crude drug were obtained through various methods of validation. The identification of the common peaks can provide a certain reference for the quality biomarkers for *Cibotii rhizoma*.

## 3. Materials and Methods

### 3.1. Materials

The dry rhizomes of *C. barometz* from the main producing area in China were purchased in February 2023. The producing areas of the samples are shown in Table 1. After being crushed and passing through a 40-mesh sieve, they were stored in the dryer.

Reference standards of protocatechuic acid (≥99%) and protocatechuic aldehyde (≥99%) were obtained from Chengdu Manster Biotechnology Co., Ltd. (Chengdu, China). Glacial acetic acid (HPLC grade) was provided by Tianjin Kermel Chemical Reagent Co., Ltd. (Tianjin, China); HPLC-grade acetonitrile was provided by Sigma-Aldrich Shanghai Trading Co., Ltd. (Shanghai, China). Pure water was obtained from China Resources C`estbon Beverage (China) Co., Ltd. (Shenzhen, China). All other reagents were analytically pure.

### 3.2. Preparation of Samples and Standard Solutions

Certain weights of reference standards were mixed with a methanol-1% acetic acid (70:30) solvent to obtain 1 mg/mL stock solutions of protocatechuic acid and protocatechualdehyde. The stock solutions were diluted to standard solutions with protocatechuic acid of 20.40 μg/mL and protocatechuic aldehyde of 9.95 μg/mL. In addition, a mixed solution of the two standards was prepared using the stock solutions.

Approximately 2 g of 40-mesh powdered sample (S1–S13) was suspended in 25 mL of methanol-1% acetic acid (70:30) by heating reflux extraction for 1 h, then cooling the mixture down and making up for weight loss with a methanol-1% acetic acid (70:30) solvent. The sample was shaken and filtered through filter paper, and the extract was collected by an evaporation solvent. The extract was redissolved with a methanol-1% acetic acid (70:30) solvent, and then transferred to a 5 mL volumetric flask. After passing the mixture through a 0.22 μm filter membrane, the sample solution was obtained. Thirteen batches of *Cibotii rhizoma* sample solutions were prepared using the above methods.

### 3.3. Apparatus and Chromatographic Conditions

A Shimadzu LC 20AT HPLC system (Shimadzu, Kyoto, Japan) equipped with a UV detector, a binary pump and an automatic sampler was used to detect soluble phenolic acid. Chromatographic separation was carried out with an Ultimate AQ-C_18_ column (4.6 mm × 250 mm, 5.0 μm, Welch, Shanghai, China) using Solvent-A (acetonitrile) and Solvent-B (1% glacial acetic acid aqueous solution). The column was maintained at 40 ℃, the injection volume was 10 μL, and the flow rate was 1.0 mL/min. Binary gradient elution was performed as follows: linear gradient 0~5% A from 0 to 20 min; 5~8% A from 20 to 45 min; 8–12% A from 45 to 85 min; maintained 12% A for five minutes (from 85 to 90 min). The detection wavelength was 260 nm. Solvent-A and Solvent-B were degassed by ultrasonic cleaners (Skymen, Shenzhen, China) for 30 min before use.

### 3.4. Method Validation

The precision test was performed using the same test solution and injecting it 6 times continuously. The stability test was performed by detecting the test solution after it was placed at room temperature for 0, 2, 4, 8, 12 and 24 h, respectively. The repeatability test was performed by detecting six sample solutions by weighing the same sample six times. The peak areas of precision, stability and repeatability were recorded separately. The precision, stability and repeatability of the method were evaluated by determining the relative standard deviation (RSD) of relative retention time (RRT) and the relative peak area (RPA) of compounds.

### 3.5. Data Analysis

Data analysis was performed using the Similarity Evaluation System for Chromatographic Fingerprint of Traditional Chinese Medicine software (Version 2012.130723). Hierarchical cluster analysis (HCA) and principal component analysis (PCA) were used for the classification of *Cibotii rhizoma* by the SPSS (Version 27.0) software, and orthogonal partial least squares discriminant analysis (OPLS-DA) was used employing the SIMCA (Version 14.1) software. HCA is an unsupervised analysis method. There is no artificial given classification label when the sample is input, and the sample data are directly modeled and classified, which can obviously distinguish the differences between groups. OPLS-DA is a supervised statistical method of discriminant analysis. It is necessary to set classification labels manually when inputting samples so as to establish a classification model. OPLS-DA can effectively distinguish the observed values between groups and find the difference variables that lead to the difference between groups.

### 3.6. LCMS-IT-TOF

LCMS analyses of the sample solution were performed on a LCMS-IT-TOF apparatus (Shimadzu, Japan) equipped with a diode array detector (SPD-M20A) and a tandem mass spectrometer with an electrospray ionization (ESI) source coupled to ion trap (IT) and time-of-flight (TOF) mass analyzers. Chromatographic conditions were used as described in Section 3.3. The water-soluble phenolic acid of *Cibotii rhizoma* was detected in negative ion mode. Chromatograms were recorded at 190 to 800 nm. The MS parameters were set as follows: detector voltage, 1.95 kV; TOF pressure, 1.9 × 10^−4^ Pa; IT pressure, 1.6 × 10^−2^ Pa; RP vacuum, 160.4 Pa; drying gas, 202.0 kPa; N_2_ flow, 1.5 L/min; CDL temperature, 200 °C; heat block temperature, 200 °C; equipment temperature, 40 °C. The full scan mass ranged from *m*/*z* 100 up to *m*/*z* 1000. The Shimadzu Composition Formula Predictor was used to predict the molecular formula. The reserpine was used as the quality calibration solution.

## 4. Conclusions

An efficient HPLC method was established for the fingerprint analysis of *Cibotii rhizoma*. Fifteen common peaks were identified, which enriched the common and different components of *Cibotii rhizoma* from different regions. Seven different components were screened out by OPLS-DA, which were P14(cibotiumbaroside G), P8(6-O-protocatechuoyl-D-glucopyranose(isomeride)), P4, P15, P5(protocatechuic acid), P12(3-caffeoyl-β-D-glucose) and P11(1-caffeoyl-β-D-glucose). The difference in the quality of *Cibotii rhizoma* was the result of the cooperation of various chemical components. Among them, protocatechuic acid was the index component stipulated in China Pharmacopeia 2020, and the unknown components P4 and P15 need to be further studied. It is recommended to consider adding other common and differential components as content measurement indicators in order to more scientifically evaluate the quality of *Cibotii rhizoma*. The similarity evaluation results of 13 batches of *Cibotii rhizoma* samples showed that the quality of *Cibotii rhizoma* from different regions was relatively stable, and the types of chemical components were basically the same, but there were great differences in the content of chemical components. As the result of HCA, the samples from Sichuan were classified into a group. Sichuan is the traditional origin of *Cibotii rhizoma* [5], and the quality of each batch was relatively stable, so it could cluster into one group. It can be seen from the literature that *Cibotii rhizoma* from Sichuan province was changed from Hebei province through the origin [26] and then changed to other places, which may be the reason for the producing area crossing of other batches. OPLS-DA can effectively distinguish the observed values between groups. The results of OPLS-DA showed that there was no cross between the three types of samples, and the distinction effect was good. Because of the quality differences in the growing environment and processing methods, the overall evaluation of the quality of the *Cibotii rhizoma* by fingerprint is conducive to further regulating the market. The main difference components screened out are potential markers of the difference in the origin of *Cibotii rhizoma*, which is expected to provide a reference for the quality control of *Cibotii rhizoma*.

The fingerprint of water-soluble phenolic acid of *Cibotii rhizoma* established in this experiment is simple, rapid and reproducible, which can reflect the quality of *Cibotii rhizoma* as a whole. A total of 15 common peaks were labeled, 2 known components were identified, 11 components were inferred by LC-MS, and *Cibotii rhizoma* was comprehensively evaluated through multiple analysis modes. Finding five known differential components and two unknown differential components that affect the quality of different regions of *Cibotii rhizoma* can provide a basis for further improvement of quality analysis, evaluation, and control of *Cibotii rhizoma*. However, there are still some unidentified components in mass spectrometry, and it is necessary to separate and identify them using various methods in order to provide a scientific basis for screening the iconic components of *Cibotii rhizoma*.

## Figures and Tables

**Figure 1 pharmaceuticals-17-00313-f001:**
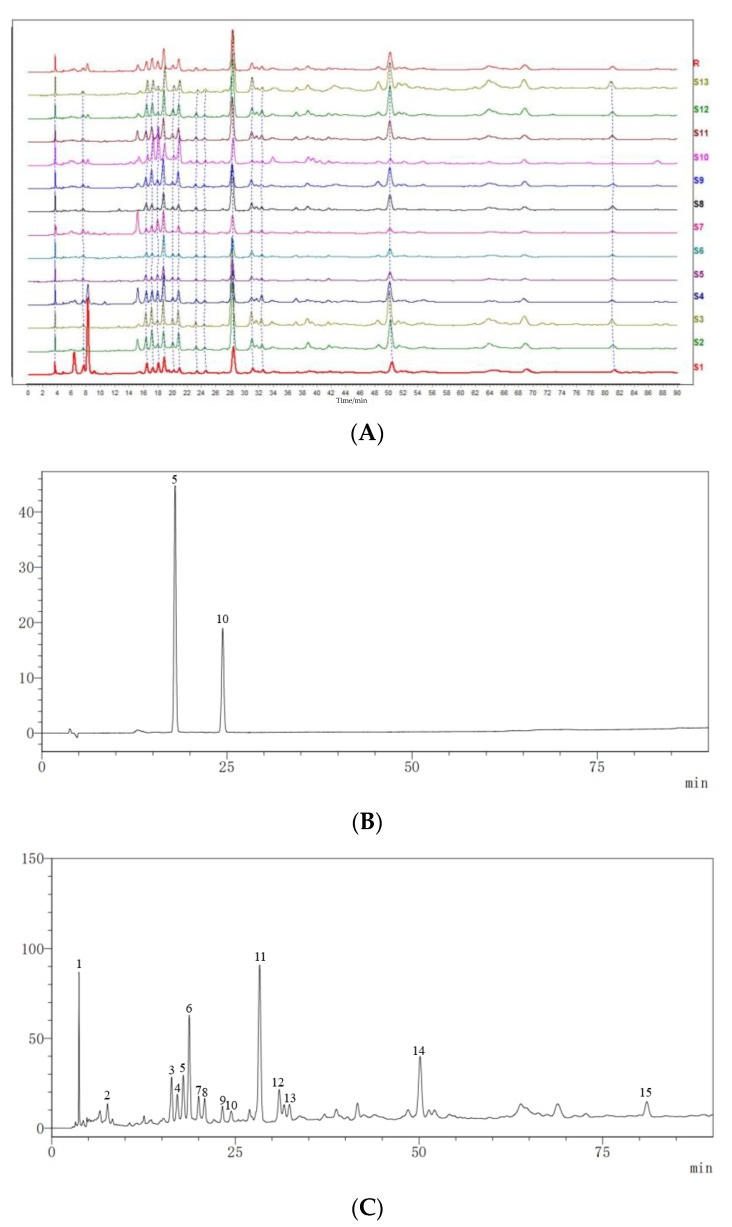
HPLC fingerprints for 13 batches of *Cibotii rhizoma* (**A**), mixed standard chromatogram (**B**) and the sample of the S5 chromatogram (**C**). (**B**): 5, protocatechuic acid, and 10, protocatechuic aldehyde.

**Figure 2 pharmaceuticals-17-00313-f002:**
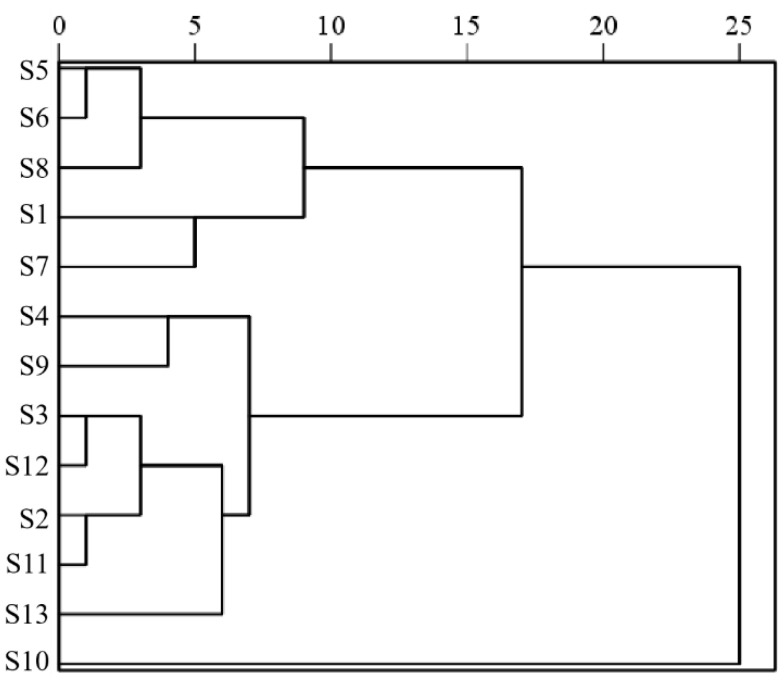
Diagram of HCA.

**Figure 3 pharmaceuticals-17-00313-f003:**
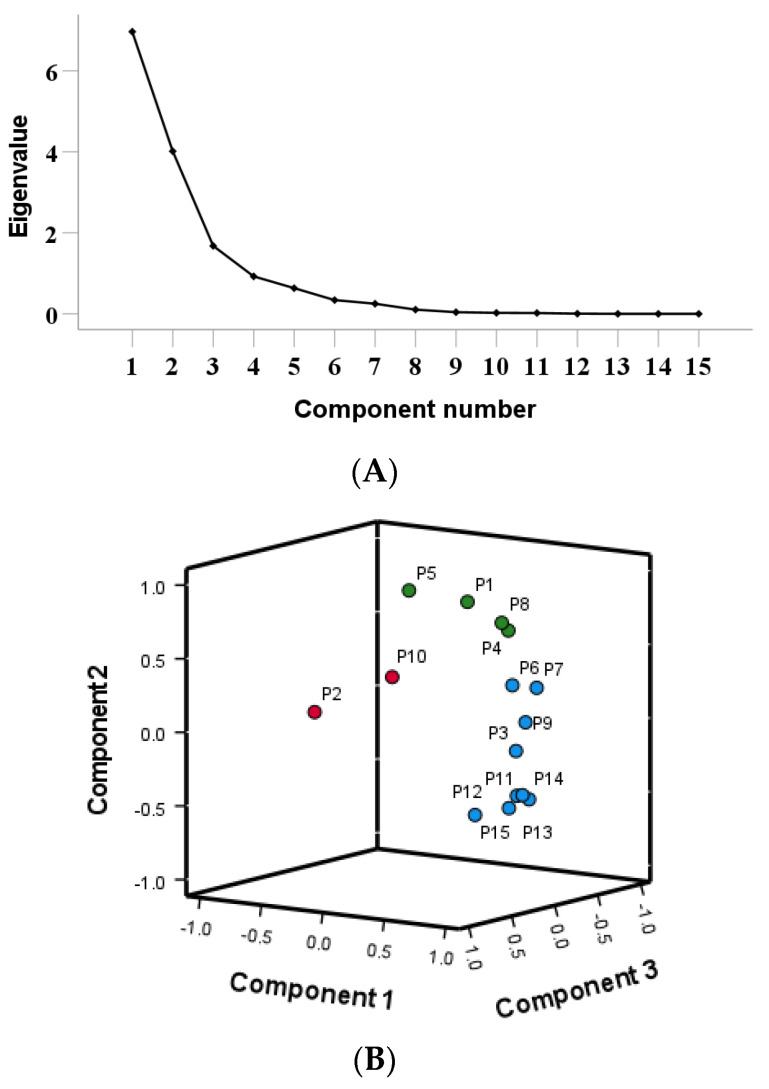
Scree plot (**A**), 3D loadings plot (**B**) and scores plot (**C**) of PCA.

**Figure 4 pharmaceuticals-17-00313-f004:**
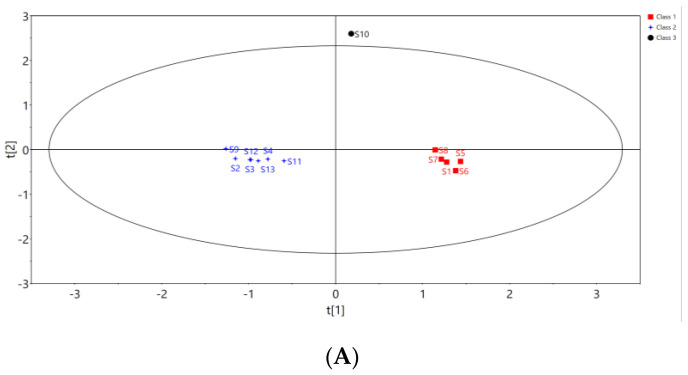
Score plot (**A**) and permutation test (**B**) of OPLS-DA, VIP values of 15 common peaks (**C**).

**Figure 5 pharmaceuticals-17-00313-f005:**
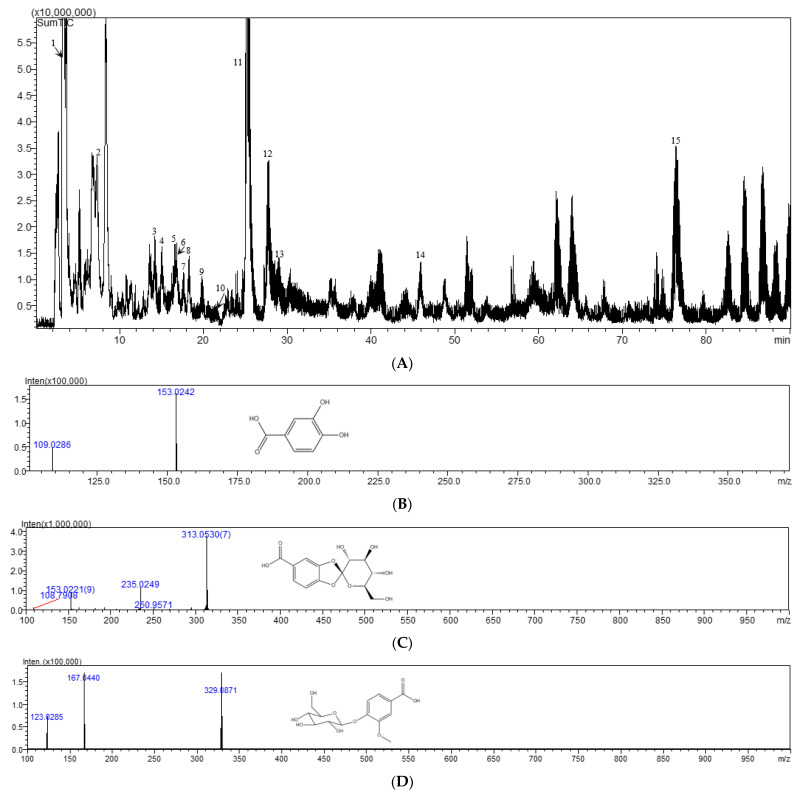
Total ion chromatogram in negative mode (**A**), MS spectra for protocatechuic acid (**B**), cyathenosin A (**C**), vanillic acid 4-O-β-D-glucopyranoside (**D**), 1-caffeoyl-β-D-glucose (**E**), and cibotiumbaroside G (**F**).

**Table 1 pharmaceuticals-17-00313-t001:** Similarity of 13 batches of samples.

Batch	Producing Area	Similarity	Batch	Producing Area	Similarity
S1	Guilin, Guangxi	0.979	S8	Baoshan, Yunnan	0.975
S2	Hezhou, Guangxi	0.981	S9	Qiandongnan, Guizhou	0.939
S3	Hezhou, Guangxi	0.975	S10	Pingxiang, Jiangxi	0.785
S4	Baise, Guangxi	0.979	S11	Longyan, Fujian	0.998
S5	Leshan, Sichuan	0.987	S12	Puning, Guangdong	0.987
S6	Meishan, Sichuan	0.944	S13	Guangzhou, Guangdong	0.989
S7	Chengdu, Sichuan	0.910			

**Table 2 pharmaceuticals-17-00313-t002:** The RPA and RSDs of 15 common chromatographic peaks of 13 samples.

	1	2	3	4	5	6	7	8	9	10	11	12	13	14	15
S1	0.120	0.253	0.305	0.194	0.340	0.496	0.130	0.183	0.078	0.084	1.000	0.199	0.124	0.500	0.156
S2	0.052	0.034	0.158	0.265	0.106	0.261	0.129	0.229	0.041	0.030	1.000	0.181	0.088	0.559	0.082
S3	0.042	0.023	0.126	0.185	0.080	0.270	0.094	0.161	0.037	0.018	1.000	0.175	0.093	0.489	0.091
S4	0.138	0.085	0.313	0.374	0.336	0.763	0.184	0.373	0.115	0.060	1.000	0.162	0.202	0.780	0.082
S5	0.190	0.088	0.242	0.168	0.267	0.556	0.168	0.146	0.076	0.060	1.000	0.223	0.123	0.507	0.137
S6	0.209	0.086	0.255	0.232	0.139	1.032	0.209	0.210	0.108	0.057	1.000	0.382	0.165	0.566	0.144
S7	0.186	0.187	0.234	0.407	0.676	1.032	0.203	0.394	0.093	0.126	1.000	0.191	0.181	0.321	0.134
S8	0.096	0.041	0.149	0.138	0.069	0.337	0.091	0.124	0.053	0.025	1.000	0.172	0.093	0.416	0.112
S9	0.191	0.089	0.380	0.704	0.331	1.076	0.268	0.632	0.105	0.084	1.000	0.336	0.109	1.001	0.140
S10	0.234	0.128	0.289	1.011	1.222	0.883	0.348	1.011	0.101	0.089	1.000	0.129	0.095	0.211	0.040
S11	0.089	0.049	0.198	0.294	0.157	0.466	0.166	0.264	0.065	0.043	1.000	0.212	0.124	0.546	0.133
S12	0.064	0.033	0.174	0.202	0.090	0.375	0.131	0.185	0.054	0.029	1.000	0.157	0.111	0.596	0.087
S13	0.069	0.045	0.163	0.193	0.127	0.413	0.123	0.212	0.039	0.041	1.000	0.273	0.095	0.564	0.190
RSD/%	51.17	76.57	33.4	75.06	106.67	49.77	41.9	78.77	38.08	54.6	0.00	34.26	29.82	35.73	33.85

**Table 3 pharmaceuticals-17-00313-t003:** The results of content determination (mg/kg).

	Protocatechuic Acid	Protocatechuic Aldehyde
S1	60.04	14.89
S2	44.23	12.50
S3	44.31	9.80
S4	69.52	12.39
S5	33.38	7.46
S6	15.82	6.50
S7	82.97	15.43
S8	20.38	7.36
S9	50.64	12.84
S10	193.81	14.18
S11	42.95	11.76
S12	36.44	11.75
S13	50.32	16.30

**Table 4 pharmaceuticals-17-00313-t004:** Eigenvalue and variance contribution rate of 13 batches of samples.

PC	Initial Eigenvalue			Extraction Sums of Squared Loading			Rotating Load Sum of Squares		
	Total	Variance/%	Accumulate/%	Total	Variance/%	Accumulate/%	Total	Variance/%	Accumulate/%
1	6.964	46.430	46.430	6.964	46.430	46.430	6.183	41.218	41.218
2	4.014	26.763	73.193	4.014	26.763	73.193	4.634	30.890	72.108
3	1.677	11.182	84.375	1.677	11.182	84.375	1.804	12.267	84.375
4	0.925	6.167	90.542						

**Table 5 pharmaceuticals-17-00313-t005:** Identification results of the chemical constituents of *Cibotii rhizoma*.

Peak *^a^*	t_R_ *^b^*/min	Tentative Identification	Molecular Formula	Measd	Predicted	Error (ppm)	Fragments
1	3.545	Quercetin methyl eter	C_16_H_14_O_7_	317.0625	317.0667	−4.2	317 [M-H]^−^
2	7.385	Mucic acid dimethyl ester gallate	C_15_H_18_O_12_	389.0736	389.0725	1.1	389 [M-H]^−^
3	14.210	6-O-protocatechuoyl-D-glucopyranose	C_13_H_16_O_9_	315.0776	315.0800	−2.4	315 [M-H]^−^, 153 [M- C_6_H_11_O_5_]^−^, 109 [M-C_6_H_11_O_5_-CO_2_]^−^
4	15.097	Unidentified	C_15_H_26_O_14_	429.1215	429.1250	−3.5	429, 315
5	16.035	Protocatechuic acid *	C_7_H_6_O_4_	153.0242	153.0193	4.9	153 [M-H]^−^, 109 [M-COOH]^−^
6	16.577	Cyathenosin A	C_13_H_14_O_9_	313.0534	313.0565	−3.1	313 [M-H]^−^, 235, 153 [M-C_6_H_9_O_5_]^−^, 109 [M-C_6_H_9_O_5_-CO_2_]^−^
7	17.595	6-O-protocatechuoyl-D-glucopyranose(isomeride)	C_13_H_16_O_9_	315.0719	315.0722	−0.3	315 [M-H]^−^, 153 [M-C_6_H_11_O_5_]^−^
8	18.363	6-O-protocatechuoyl-D-glucopyranose(isomeride)	C_13_H_16_O_9_	315.0718	315.0722	−0.4	631 [2M-H]^−^, 315 [M-H]^−^, 255
9	20.417	Vanillic acid 4-β-D-glucopyranoside	C_14_H_18_O_9_	329.0881	329.0878	0.3	329 [M-H]^−^, 167 [M-C_6_H_11_O_5_]^−^, 123 [M-C_6_H_11_O_5_- CO_2_]^−^
10	22.163	Protocatechuic aldehyde *	C_7_H_6_O_3_	137.0220	137.0244	−2.4	137 [M-H]^−^
11	25.857	1-caffeoyl-β-D-glucose	C_15_H_18_O_9_	341.0865	341.0878	−1.3	341 [M-H]^−^, 281 [M-COOH]^−^, 233, 179 [M-COOH-C_6_H_11_O_5_]^−^
12	27.363	3-caffeoyl-β-D-glucose	C_15_H_18_O_9_	341.0893	341.0878	1.5	341 [M-H]^−^, 281 [M-COOH]^−^, 233, 203, 179 [M-COOH-C_6_H_11_O_5_]
13	28.957	4-caffeoyl-β-D-glucose	C_15_H_18_O_9_	341.0859	341.0878	−1.9	341 [M-H]^−^, 281 [M-COOH]^−^, 203, 179 [M-COOH-C_6_H_11_O_5_]
14	45.730	Cibotiumbaroside G	C_18_H_20_O_11_	411.0959	411.0933	2.6	411 [M-H]^−^, 315 [M-C_5_H_5_O_2_]^−^, 249
15	76.873	Unidentified	C_20_H_26_O_10_	425.1443	425.1453	−1.0	425, 395

‘*a*’ The peak numbers are according to the peak labels in Figure 1. ‘*b*’ The retention times are according to the HPLC-ESI (-)-MS. ‘*’ comparison with the reference substance.

## Data Availability

Data is contained within the article.

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
