# Peer review of "HPLC Fingerprint Analysis of Cibotii rhizoma from Different Regions and Identification of Common Peaks by LC-MS"

_pharmaceuticals, 2024, doi:10.3390/ph17030313_

Round 1
Reviewer 1 Report
Comments and Suggestions for Authors
This is a study on the chromatographic analysis of various Cibotii Rhizoma samples. The general draft is good, but the work would be better if some additions were made.
The names of the 2 components analyzed and identified should be given in the abstract section (protocatechuic acid and protocatechuic aldehyde).
Examples from other previous studies on this subject should be given in the discussion section. A section stating the advantage of this study over other studies should be added. What makes this study unique from other studies?
It has been stated that there are very few studies on this subject in the literature. If possible, can the amounts of the determined compounds in mg/kg be given? These results will contribute to the literature (at least for these two standard species: protocatechuic acid and protocatechuic aldehyde).
A short paragraph can be added explaining the importance of these two phenolic compounds.
​
Author Response
Dear reviewer:
We feel great thanks for your professional review work on my manuscript. As you are concerned, there are several problems that need to be addressed. We have tried our best to improve and made some changes in the manuscript. The yellow part that has been revised according to your comments. The detailed corrections are listed below:
1.The names of the 2 components analyzed and identified should be given in the abstract section (protocatechuic acid and protocatechuic aldehyde).
We sincerely thank the reviewer for careful reading. As suggested by the reviewer, we have given the name of the 2 components in the abstract section.
2.Examples from other previous studies on this subject should be given in the discussion section. A section stating the advantage of this study over other studies should be added. What makes this study unique from other studies?
We tried our best to improve the manuscript and added a section 2.9 to the manuscript. These changes will not influence the content and framework of the paper. And here we did not list the changes but marked in yellow in the revised paper. We appreciate for Editors/Reviewers’ warm work earnestly and hope that the correction will meet with approval.
3.It has been stated that there are very few studies on this subject in the literature. If possible, can the amounts of the determined compounds in mg/kg be given? These results will contribute to the literature (at least for these two standard species: protocatechuic acid and protocatechuic aldehyde).
Thanks for your careful checks. We have given the amounts of the two standard species in Table 3.
4.A short paragraph can be added explaining the importance of these two phenolic compounds.
We sincerely thank the reviewer for careful reading. As suggested by the reviewer, we have added a short paragraph to explain the importance of these two phenolic compounds. The changes marked in yellow in the revised paper.
Thank you again for your positive comments and valuable suggestions to improve the quality of our manuscript.

Reviewer 2 Report
Comments and Suggestions for Authors
Authors Zhongjin Guo, Yu Duan, Zhimin Zhao, Depo Yang, Xinjun Xu send for revision their manuscript entitled: HLC fingerprint analysis of Cibotii Rhizoma from different regions and dentification of common peaks by LC-MS ‘ to the journal Pharmaceuticals.
Manuscript needs some revision and part that sholud be improved are listed below:
1. Please remove this part from Introduction section, (these details need to present in Material and Methods section): ‘ The chromatography was performed on an Ultimate AQ-C18 column with mobile phase consisting of methanol (A) -1% glacial acetic acid solution (B).
2. Page 3, part : 2.3. Establishment of HPLC fingerprint ”software of Similarity Evaluation System for Chromatographic Fingerprint of Traditional Chinese Medicine.” Please add some refernces to this part.
3. Fig. 1 Why chromatogram of sample S5 was choosen and what does maen. Please add some explanation or add reference to other part of manuscript. Maybe Authors can add in this sentence instead of : ‘ Samples were injected’ - ‘Samples S1-S13 were injected
4. Page 4: Why Authors did not add chromatofram of S10 in case it quite different from others 12 samples.
5. Poage 8-9, In Fig.3 image A. please resize captions
6. Page 12 Materials and Method section there were perpared 13 solutions?
In introduction section there was mentioned: „Thirteen batches of Cibotii Rhizoma samples were analyzed” but in the materisls and Method section there were no written that 13 solutions were prepared and how they were extracted from plants matherials.
Comments on the Quality of English LanguageAuthors Zhongjin Guo, Yu Duan, Zhimin Zhao, Depo Yang, Xinjun Xu send for revision their manuscript entitled: HLC fingerprint analysis of Cibotii Rhizoma from different regions and dentification of common peaks by LC-MS ‘ to the journal Pharmaceuticals.
Manuscript needs some revision and part that sholud be improved are listed below:
1. Please remove this part from Introduction section, (these details need to present in Material and Methods section): ‘ The chromatography was performed on an Ultimate AQ-C18 column with mobile phase consisting of methanol (A) -1% glacial acetic acid solution (B).
2. Page 3, part : 2.3. Establishment of HPLC fingerprint ”software of Similarity Evaluation System for Chromatographic Fingerprint of Traditional Chinese Medicine.” Please add some refernces to this part.
3. Fig. 1 Why chromatogram of sample S5 was choosen and what does maen. Please add some explanation or add reference to other part of manuscript. Maybe Authors can add in this sentence instead of : ‘ Samples were injected’ - ‘Samples S1-S13 were injected
4. Page 4: Why Authors did not add chromatofram of S10 in case it quite different from others 12 samples.
5. Poage 8-9, In Fig.3 image A. please resize captions
6. Page 12 Materials and Method section there were perpared 13 solutions?
In introduction section there was mentioned: „Thirteen batches of Cibotii Rhizoma samples were analyzed” but in the materisls and Method section there were no written that 13 solutions were prepared and how they were extracted from plants matherials.
